# Identifying Hydraulic Characteristics Related to Fishery Activities Using Numerical Analysis and an Automatic Identification System of a Fishing Vessel

Sung-Chul Jang [1] , Jin-Yong Jeong [2], Seung-Woo Lee [2] and Dongha Kim [2,*]

1 Research Center for Ocean Industrial and Development, Pukyong National University, Busan 48513, Korea
2 Marine Disaster Research Center, Korea Institute of Ocean Science and Technology, Busan 49111, Korea
* Correspondence: kimdh@kiost.ac.kr; Tel.: +82-51-664-3711

**Abstract:** Many countries worldwide promote artificial reef projects to increase and preserve fishery resources; however, how artificial reefs form fisheries is unclear. Nevertheless, specific hydraulic features of artificial reefs may attract fish. We selected an underwater reef as a research site to clarify this hypothesis. In this study, environmental conditions around the underwater reef were modeled and quantitatively assessed using numerical analysis. We identified two hydraulic features related to fish attraction: the wake region and the local upwelling region. Their spatial distributions were superimposed on the path of a fishing vessel that was monitored using an automatic identification system (AIS). We showed that various hydraulic characteristics (such as wake region, local upwelling region, and flow velocity) identified in the path of the fishing vessel can be quantitatively evaluated. Increasing amounts of information from the AIS can be used to identify the hydraulic features that attract the most fish and therefore improve the productivities of artificial reefs.

**Keywords:** fish activity; fish attraction; fishing vessel; numerical analysis; automatic identification system; underwater reef; wake region; local upwelling region; Ieodo ocean research station

## 1. Introduction

The "attraction and production" debate belabors whether artificial reefs (ARs) primarily attract fish or spawn the production of new fish biomass [1]. Figure 1 illustrates the marine ecosystem according to each opinion. The natural state initially features two biomasses (Figure 1a). According to those who favor attraction, the amount of biomass does not change—only its spatial distribution (Figure 1b). On the other hand, those who support the production theory believe that the magnitude of the total biomass increases because the $Q_{AR}$ in the AR generates additional biomass (Figure 1c). Which opinion is correct remains unclear, but studies have found that ARs include either or both attraction and production effects [2–7]. Overall, this debate has one prerequisite: the attraction effect generates an ecosystem in the AR. However, the mechanisms that cause the attraction effect are unknown [8]. Some studies have identified specific hydraulic characteristics that may attract fish, but these candidates may not align with the spatial distribution of marine life. Elucidating these hydraulic features would enable more effective designs of ARs.

Biological hotspot waters can be used to evaluate fish attraction. For example, submarine topography, such as a seamount (or an underwater reef), induces several hydrodynamic features such as turbulence, wake region, internal waves, and upwelling that occur in an uplifted seabed or valley and affect the abundance of fish resources and biodiversity [9–13]. One study categorized the topography of the seabed into open slopes, deep basins, canyons, and seamounts, whose biodiversity patterns differed [14]. Upwelling benefits a seamount because the current can flow along the slopes of the seamount, and turbulence diffuses nutrients and minerals. In addition, the wake region is known to have the effect of attracting marine life [15–17]. Gove et al. [9] clarified that hydrodynamic properties (upwelling,

mixing effects, internal waves, etc.) as well as the inflow of matter (including byproducts of human activities) help form biological hotspots.

Here, an underwater reef with high fishing intensity in the offshore was studied to reveal which hydraulic characteristics attract fish. Geometrically asymmetric underwater reefs exhibit different hydraulic characteristics in all current directions; therefore, we compared the different characteristics that occur around the underwater reef in all the current directions. In addition, the spatial distribution of marine life was predicted by tracing the path of a fishing vessel (FV) in search of a fishing ground and superimposing information obtained from an automatic identification system (AIS) to evaluate the hydraulic features related to the location of the fishery.

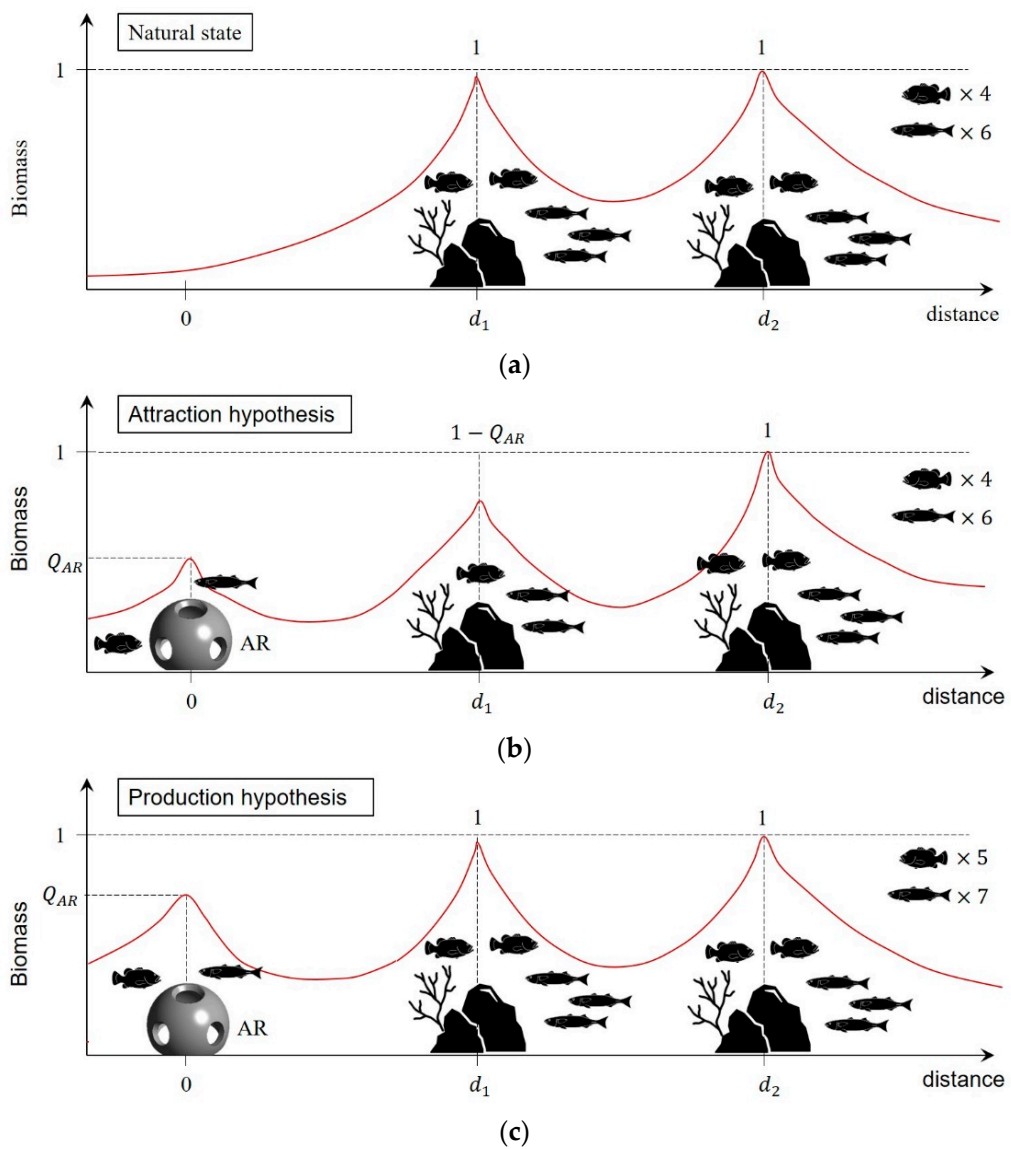

**Figure 1.** The attraction and production debate: (**a**) Initially there are two biomasses in a natural reef; (**b**) the attraction hypothesis: the movement of biomass changes after an AR is installed, which attracts some biomass ($Q_{AR}$) from the nearby natural reef ($d_1$). However, the total biomass does not change to 2; (**c**) The production hypothesis: ARs increase total biomass ($2 + Q_{AR}$) because the AR serves as a spawning ground.

## 2. Materials and Methods

This study was divided into two phases (Figure 2): locating the fishery [18] and a numerical analysis to quantitatively characterize the fluid in the target waters. First, the

location of the FV was accurately found from the AIS information, and the position of the fishery was estimated by assuming that the FV moved to the location of the fishery. In addition, the current direction through the numerical tidal map was determined from the time information of the AIS according to the position of the FV. Second, the topography of the seabed, water depth, and current direction were considered during preprocessing; the candidates expected to be related to fish attraction were selected and quantified according to the current direction. The relationship between fish attraction and hydraulic characteristics was assessed by superimposing the numerical analysis result on the path of the FV generated from the AIS.

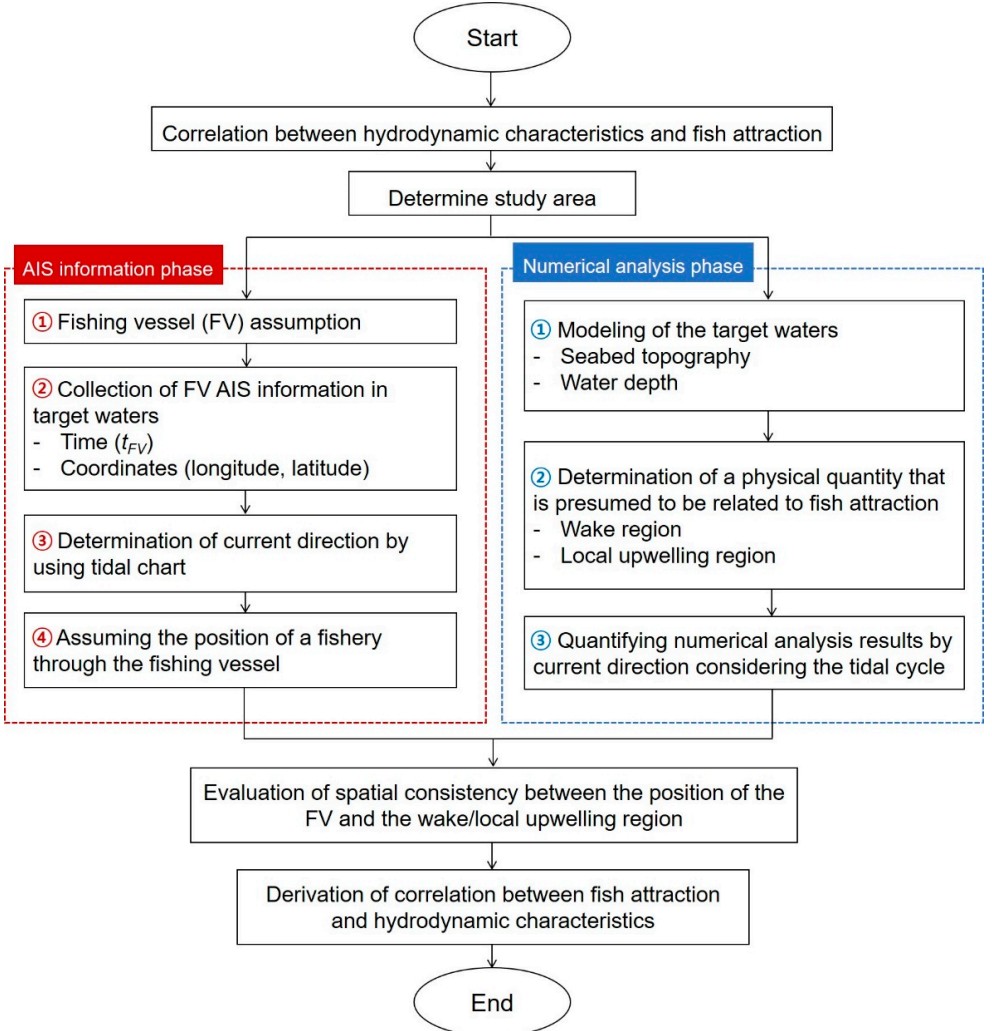

**Figure 2.** Research flowchart of this study.

## 2.1. The Target Waters and Underwater Reef

The target underwater reef, called Ieodo (or Socotra Rock), is located at the southernmost tip of Jeju Island (see Figure 3a). Its most shallow peak is about 7.1 m (based on approximately highest high water). Ieodo ranges about 1700 m north–south and 1200 m east–west based on a water depth of 47 m; it slopes steeply in southern and eastern regions and gently in northern and western regions (Figure 3b). The Ieodo Ocean Research Station (I-ORS) was constructed on the rock to better cope with natural disasters (e.g., earthquakes, typhoons, and climate change) and understand the ocean dynamics in the East China Sea (ECS) [19,20].

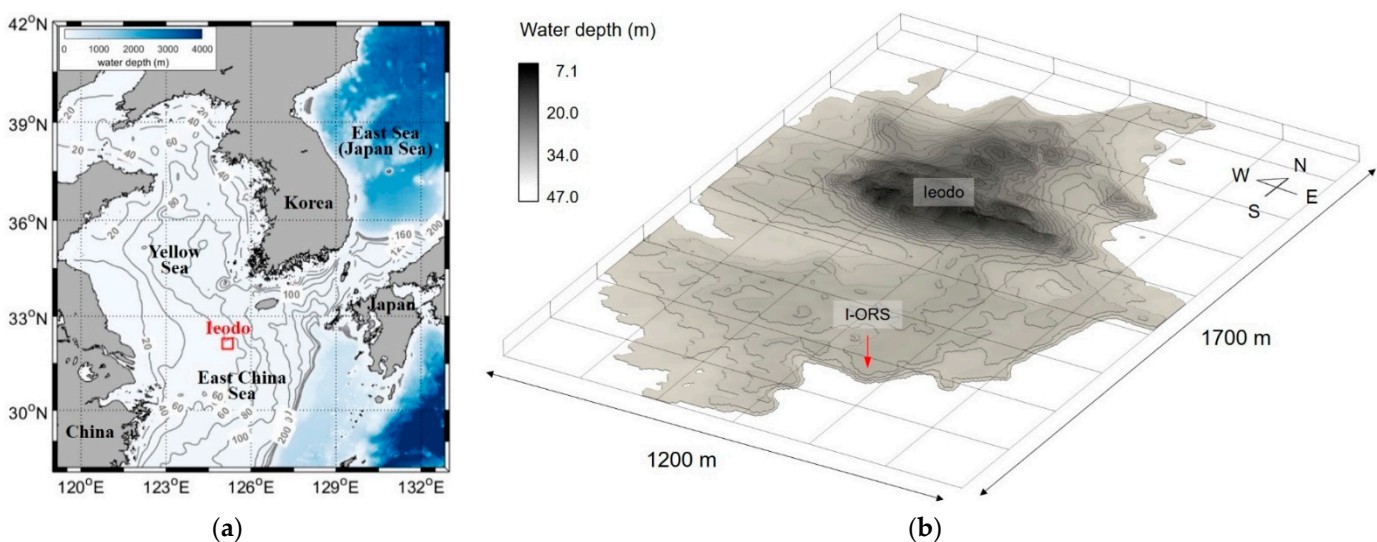

**Figure 3.** Study area and the target underwater reef (Ieodo): (**a**) Location of the study area (Ieodo); (**b**) Topography of the underwater reef. Its top is located at a depth of 7.1 m, and its southern slope from the top is very steep.

### *2.2. AIS Information Phase*

#### 2.2.1. FV Assumption

A fish finder can reliably find a fishery. The location of the fishery can, however, be indirectly estimated by assuming that there is always a fish finder on a FV of a prerequisite size. This is because the FV will inevitably encounter the fishery where fish gather. In this study, this is called the FV assumption, which presumes that the location of the fishery matches that of the FV.

#### 2.2.2. FV Tracking Using Automatic Identification System Information

AIS is a navigational tool that can receive information on land control system while transmitting information such as its location to other nearby vessels through a wireless data communication system. In this study, AIS was used to track the FV passing through the target waters. The AIS information included IMO number (MMSI number), vessel type, speed, heading, latitude, longitude, and time information. The FV was identified from the vessel information, the fishing activity was judged from the speed information, and the position of the FV was confirmed from the latitude and longitude. Finally, the tide-induced direction of the current was determined from the time information of the AIS and the numerical tidal current map, the Ocean Data in Grid Framework provided by Korea Hydrographic and Oceanographic Agency (KHOA) [21]. This system updated the numerical tidal current map every 10 min.

### *2.3. Numerical Analysis Phase*

#### 2.3.1. Framework of Numerical Analysis

In this study, commercial code FLOW-3D was used as a numerical model to estimate the hydraulic characteristics of the waters around the underwater reef (Ieodo) caused by the current directions. Each element was numerically calculated using the finite volume method, and the turbulent flow energy ($k$) and turbulent dissipation rate ($\varepsilon$) were modeled by their respective transport equations. The volume of fluid (VOF) method and fractional area volume obstacle representation were also applied to calculate the free surface of the domain and the hydraulic characteristics of the underwater topography and at the boundary. In these methods, the surface and volume of parts blocked by obstacles were calculated during preprocessing to construct an efficient numerical analysis model.



The analysis program (FLOW-3D) used a Eulerian–Eulerian approach to simulate flow and compute equations explicitly and implicitly. The turbulence region was solved based on the continuity equation (Equation (1)) and the equation of momentum (Equation (2)); the effect of sudden fluctuations was minimized. The boundary and free surface flow of each fluid were reproduced using the VOF method. The continuity and momentum equations for incompressible fluids including VOF variables are expressed as follows:

$$\frac{\partial}{\partial x_i}(u_i A_i) = 0 \tag{1}$$

$$\frac{\partial u_i}{\partial t} + \frac{1}{V_F}\left(u_j A_j \frac{\partial u_i}{\partial x_i}\right) = -\frac{1}{\rho}\frac{\partial p}{\partial x_i} + \frac{\partial}{p\partial x_i}\left(\mu \frac{\partial u_i}{x_j} - \overline{u_i' u_j'}\right) + g_i \tag{2}$$

Here, $u_i$ and $u_i'$ are average velocity and velocity fluctuation, $A_i$ is the fractional area open to flow in the directions of the element, $V_F$ is the volume in contact with the fluid, and $p$ and $g_i$ are pressures and weight forces, respectively, for the three perpendicular directions in Cartesian coordinates ($i$ = 1, 2, and 3). A turbulence model is required to simulate the Reynolds stress expression ($-\overline{u_i' u_j'}$) in Equation (2).

The area around the top of Ieodo was modeled to investigate changes in currents due to underwater reefs using numerical model simulation. The size of the domain was 1200 × 1700 × 50 m, and its topography was modeled with a resolution of 10 m. The fluid in the simulation was designated as water with a density of 1024 kg m$^{-3}$ and a viscosity of $1.0 \times 10^{-3}$ kg m$^{-1}$s$^{-1}$. In this simulation, the maximum aspect ratio was 1.05, and variable elements were configured to prevent numerical errors at the side boundary (inlet condition). The number of elements in the calculation domain was $5.83 \times 10^6$. A variation of less than $5.0 \times 10^{-4}$ from the mean was considered steady state in this simulation because unsteady state flow in a turbulent state occurs as an initial flow. The boundary conditions of the domain were assigned as follows: its wall boundary at the bottom obeyed the no-slip condition as well as zero velocity condition normal to the boundary, the top boundary included a static pressure of 1 atm, and the side boundaries included the inlet and the outlet according to the direction of each current, whose flow velocity was considered to be 1 ms$^{-1}$.

### 2.3.2. Identification of Hydraulic Features Related to Fish Attraction

Two hydraulic features related to fish attraction were determined as follows: First, the wake region was defined as a region whose current direction was reversed relative to the main direction (inflow direction) as shown in Figure 4a. The wake region occurs mainly at the back of a blunt body such as a concrete structure that blocks the current [22]. The magnitude of current velocity in the wake region is 20–30% of the surrounding current velocity [22].

Second, we considered upwelling—a phenomenon in which low-temperature, nutrient-rich deep water rises to the surface layer in a high-temperature, nutrient-depleted state. Good fisheries form in waters where upwelling occurs due to abundant nutrients, and high fishery production can be expected. Local upwelling was induced so that the current could climb up the artificial structure. Biomass and biodiversity increased rapidly over 10 months [23], indicating that artificial structures can improve marine productivity. We therefore used a numerical model based on high-resolution seabed topography data to analyze local upwelling.

In this study, an upward current in target waters (1.7 × 1.2 km) was defined as a "local upwelling region" and analyzed using a numerical model (see Figure 4b). The seabed topography of the target waters was analyzed with a resolution of 10 m, and the spatial distribution and volume of the local upwelling region were quantified according to water depth and current direction. The local upwelling region was defined as a region with a vertical velocity component ($w$) greater than 5% of the inlet current velocity; the value of 5% classifies the local upwelling region that occurs according to the current directions.

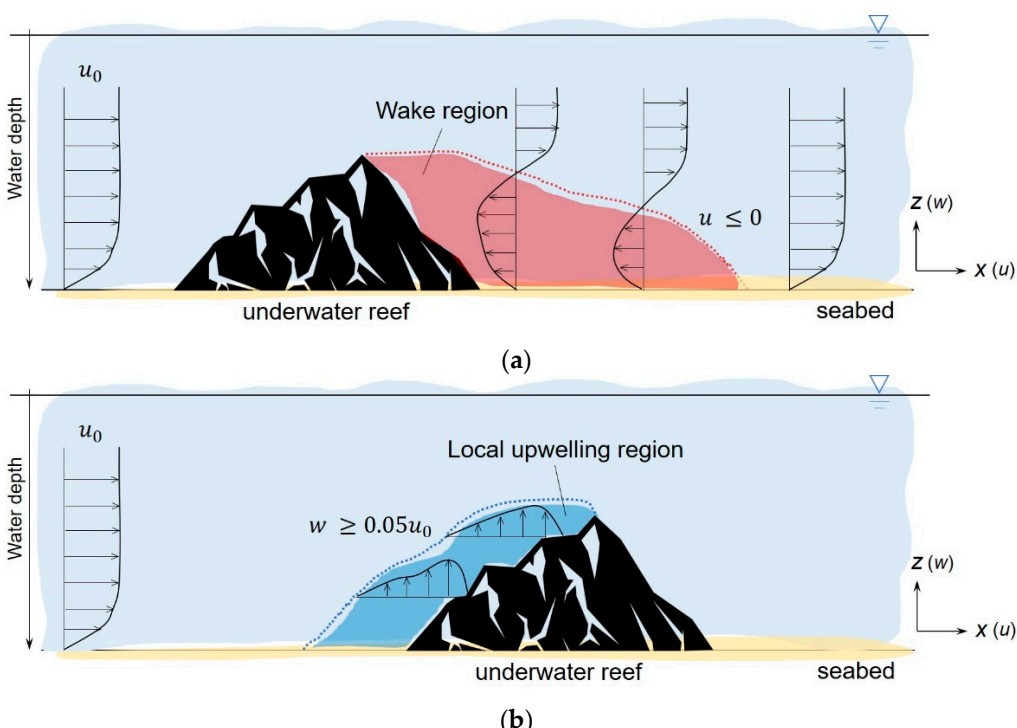

**Figure 4.** Hydrodynamic features that can render an underwater reef a biological hotspot: (**a**) The wake region (red part) was defined as a region with negative $u$-velocity. It primarily occurs downstream of an underwater reef; (**b**) the local upwelling region (blue part) was defined as the $w$-velocity region greater than 5% of the inlet current velocity ($u_0$). It occurs primarily on upstream slopes due to the uplifted seabed.

## 3. Results

### 3.1. Quantification of Two Candidates Related Fish Attraction Using Mumerical Analysis

We used numerical analysis to quantitatively characterize wake regions and local upwelling regions, whose hydraulic characteristics may relate to fish attraction. Figure 5 shows the results for 12 directions at intervals of $30°$ corresponding to the direction of the current, which changes according to the tidal cycle. The geometric asymmetry of the underwater reef greatly influences hydraulic characteristics depending on the current direction. The wake region had the greatest volume when the current direction was is $0°$: $1,645,191\ \text{m}^3$. Spatially, a large amount of wake region arose on the southern steep cliff; this pattern has been noted in previous numerical analyses [24–28]. The wake region usually occurs in conditions where an object blocks the flow. This is because the wake region occurs from pressure gradients and turbulence downstream of the object. This pattern can also be found in previous studies [10,29,30]. In the case of Ieodo, since the slope in the south direction is very steep, there is a large wake region in the $0°$ direction. On the other hand, in the $180°$ direction, the wake region is relatively small compared with $0°$. This is because the terrain changes at the north of the island is gentle. The local upwelling region is governed by the slope of the seabed topography. A large-scale local upwelling region develops when it flows along a steep slope of the seabed topography. Therefore, there is a characteristic that mainly appears upstream from the top of the Ieodo. As shown in Figure A1 (included in the Appendix A), it can be seen that the location where the local upwelling region and the wake region appear is spatially separated. However, a wake region did not emerge in the $120°$ direction because the topography did not change drastically enough to create one. The deviation of the wake region's occurrence was large.

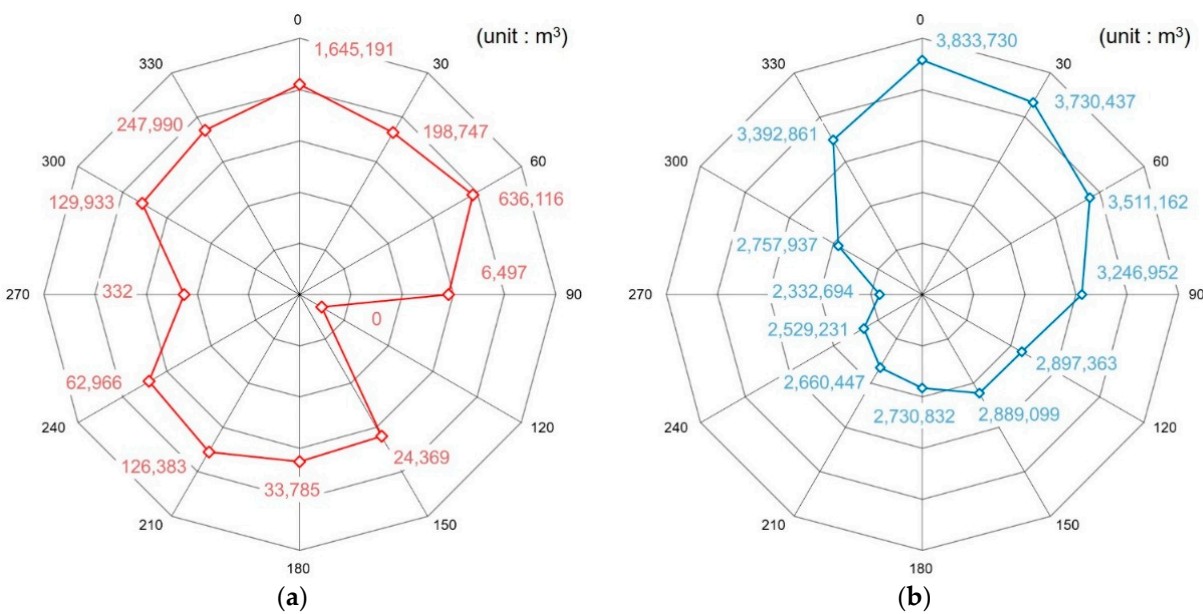

**Figure 5.** Volume diagrams of the wake region volume and the local upwelling region, which are shown according to the current direction from the numerical analysis: (**a**) wake region volume diagram (expressed in log scale); (**b**) local upwelling volume diagram.

The largest volume of the local upwelling region was 3,833,730 m$^3$ when the current direction was 0°, and the lowest volume was 2,332,694 m$^3$ when the current direction was 270°. However, unlike the wake region, its deviation of the volume according to current direction is relatively small. Moreover, the local upwelling region primarily occurred on the upstream slope of Ieodo. Figure 6 shows the maxima and minima of the wake region and local upwelling region, which mostly occurred when the current direction was 0°. On the other hand, the minimum volumes of the wake region and local upwelling region occurred at 120° and 270°, respectively.

Figure A2 (included in the Appendix A) shows the vertical velocity distribution in the flow direction passing through the top of Ieodo to investigate the change of local upwelling according to the flow direction. In the results, it was confirmed that the magnitude of the vertical velocity was strongly shown in front of the top, and in particular, it was found to be high at 120°, 150°, 180°, and 210° which had a steep. This is related to the thick distribution of local upwelling around the top in Figure A1. Additionally, at 0°, 30° which has a steep slope behind the top, the local upwelling occurred behind the top, which is thought to be caused by inducing circulation flow due to the creation of the wake region. This indirectly confirms that the slope behind top is related to the formation of the wake region.

### 3.2. Superimposition of AIS Information-Based FV Path and Numerical Analysis Results

The path of FV1 was superimposed on the numerical analysis results in the direction of the current at that time (Figure 7) by using the AIS-based location and time information of FV1. The FV1's AIS information was collected on 11 March 2022 from 19:41–23:23 (GMT). A white dot indicates the position recorded when the AIS information was collected. Table 1 shows the AIS information of FV1. At this time, the current direction could be determined through a numerical tidal current map. Thus, it can be checked whether there is a path of FV1 at the location where the wake region and the local upwelling region occur. As FV1 moved, the direction changed to 120°, 150°, and 180° as shown in Figure 7a, b, and c, respectively. FV1 could be used to confirm where the local upwelling region occurred. The local upwelling region that arose in the northern region of Ieodo (P4–7 at Figure 7) changed to a wake region when the current direction shifted from 150° to 180°. Likewise, the path of FV1 was concentrated in the northern region of Ieodo.

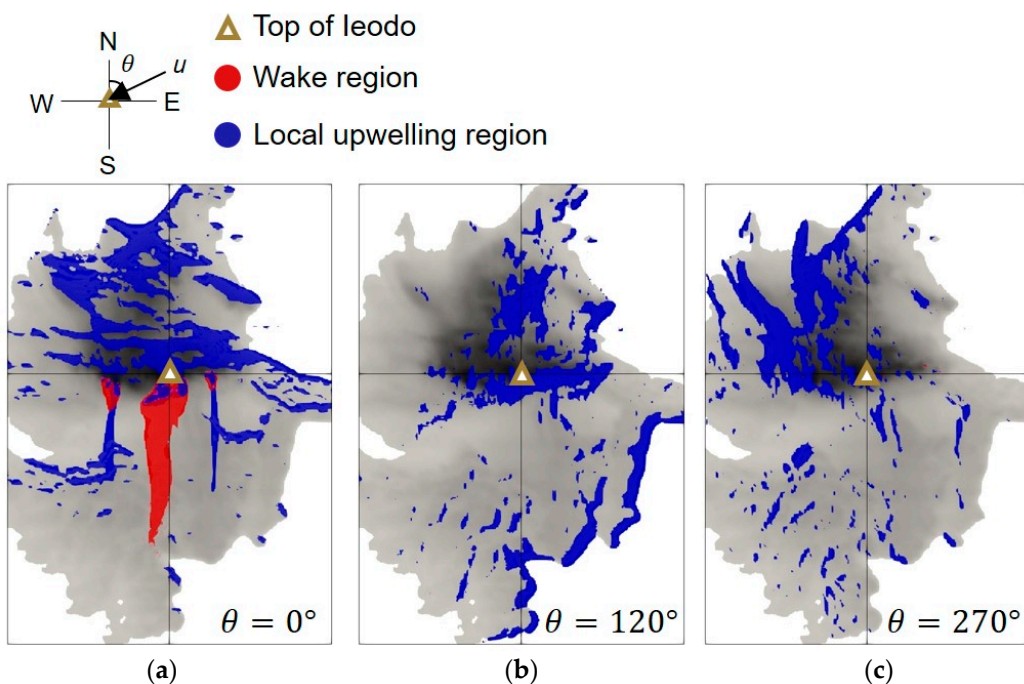

**Figure 6.** Spatial distribution and quantitative profiling of the wake region and local upwelling region in the target waters: (**a**) The volumes of both the wake region and local upwelling region were greatest when the direction of the current was 0°; (**b**) the wake region did not appear in the 120° direction of the current, whereas the local upwelling region developed along the slope; (**c**) the volume of the local upwelling region minimized at 270°.

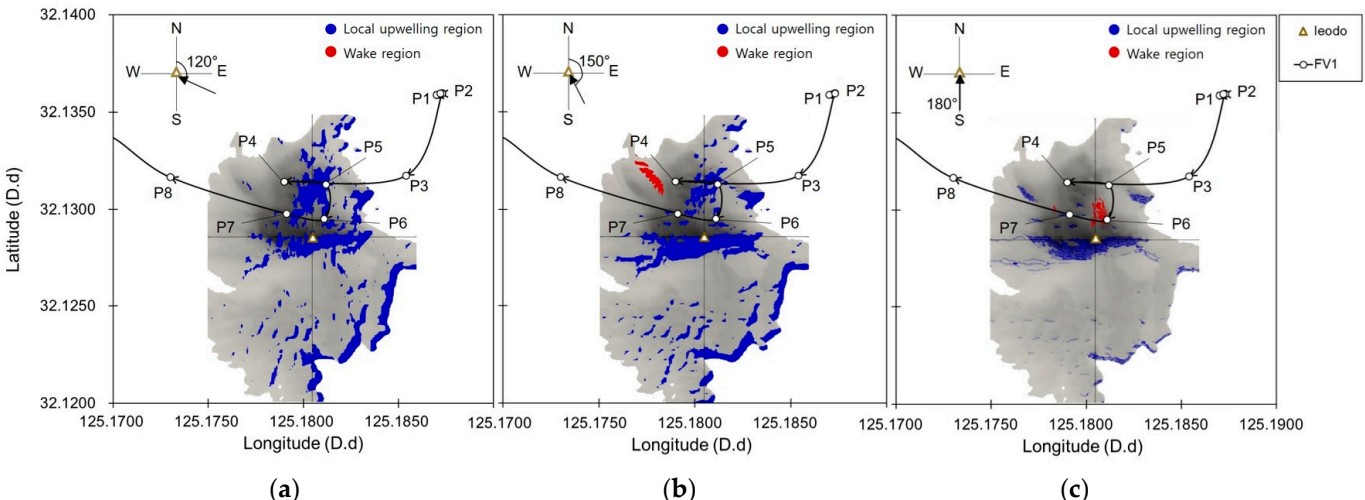

**Figure 7.** The spatial distribution of the wake region and local upwelling region around the underwater reef according to numerical analysis. A solid line indicates the path of FV1. The AIS information was collected for 3 h, during which time the current direction changed from 120° to 180°: (**a**) Numerical analysis result when the current direction was 120° from (GMT) 19:41 to 20:40 on 11 March 2022; (**b**) current direction is 150° from 20:41 to 21:40; (**c**) current direction is 180° from 21:41 to 22:40.

**Table 1.** Location and time of FV1 obtained from AIS information.

| Code | LON | LAT | Timestamp (GMT) | Location |
|------|-----|-----|-----------------|----------|
| | 125.1871 | 32.13589 | 11 March 2022 19:41 | P1 |
| | 125.1873 | 32.13599 | 11 March 2022 21:53 | P2 |
| | 125.1854 | 32.13173 | 11 March 2022 22:11 | P3 |
| | 125.1790 | 32.13143 | 11 March 2022 22:13 | P4 |
| FV1 | 125.1812 | 32.13128 | 11 March 2022 22:21 | P5 |
| | 125.1811 | 32.12952 | 11 March 2022 22:23 | P6 |
| | 125.1791 | 32.12979 | 11 March 2022 22:25 | P7 |
| | 125.1730 | 32.13166 | 11 March 2022 22:40 | P8 |

## 4. Discussion

In this study, the wake region occurring in the actual seabed topography was numerically analyzed. In addition, the volume and spatial distribution of the wake region were used to determine whether the wake region correlated with fish attraction according to the current direction, the tidal cycle, and the location of the FV. The upwelling region accounts for 5% of the total ocean and 25% of the world's total number of fish [31]. In 1991, Japan attempted to build an artificial structure with a width of 45 m and a height of 10 m to profit from upwelling and to increase catches [32]. Upwelling occurs primarily on the coast, which is explained by wind-driven Ekman transport. However, it is almost impossible to observe upwelling because its vertical velocity is very slow. For example, a previous numerical model of the coast of Oregon state, USA determined the upwelling velocity to be $2 \times 10^{-2}$ cm s$^{-1}$ [33]. Upwelling that occurs locally along the uplift seabed is especially difficult to predict or observe because its mechanism is unrelated to Ekman transport. The wake region helps to form an artificial fishery by providing a shelter and spawning ground for fish [15,17,34–38]. These studies showed that an artificial fishery is more likely to form in larger wake regions with lower internal currents. If successful, the spatial correlation between the wake region and artificial fishery can be determined. Good fisheries most commonly form in wake regions around seabed uplifts such as underwater reefs, where FVs spontaneously gather.

Hydraulic features that explain fish attraction can be identified and quantified from numerical analysis if the FV assumption is valid. Better AR projects can be developed by including these hydraulic features that attract fish. Currently, countries that operate ARs are concerned about how their shape and placement can better attract fish; accordingly, an AR is recorded on video shortly after installation to monitor the gathering of marine life. Although the ecological habits of most marine life are often unknown, our study identified factors that attract fish from a macroscopic perspective.

Our numerical analyses and the AIS information on FV1 show that the local upwelling region correlates with high-quality fisheries, which agrees with previous studies [23,37,38]. However, this study has some limitations: First, we obtained relatively little information from the AIS to track FV because large amounts are not supported statistically. More information could better clarify the relationship between fish attraction, wake regions, and local upwelling regions. Second, the FV assumption is uncertain but can be modified according to the FV's target marine life species, fishing gear, fishing methods, and the captain's control. Finally, the numerical analysis may be unreliable because it is difficult to accurately predict fluid behavior using the numerical model. Time and space were approximated in our quantitative numerical analyses. Thus, it is necessary to sufficiently converge the error within the target error range.

## 5. Conclusions

In this study, an AIS was used to track an FV, and numerical analysis revealed and quantitatively evaluated hydraulic features that attract fish: the wake region and the local upwelling region. Fish attraction was assessed by superimposing the numerical analysis result and information from the AIS used to track the FV. We showed that a reliable

numerical model can predict the hydraulic features in a specific aquatic region if sufficient quantitative information is collected from an AIS. The FV assumption is valid if fishing activity is concentrated in a relatively small region; this study will inform future numerical models that can be used to develop highly productive ARs worldwide.

**Author Contributions:** Conceptualization, D.K. and S.-C.J.; methodology, D.K. and S.-W.L.; software, D.K. and S.-C.J.; validation, D.K. and S.-W.L.; formal analysis, S.-C.J.; investigation, D.K. and S.-C.J.; data curation, S.-C.J. and S.-W.L.; writing—original draft preparation, S.-C.J.; writing—review and editing, D.K., S.-W.L. and J.-Y.J.; visualization, D.K. and S.-C.J.; supervision, J.-Y.J.; project administration, J.-Y.J.; funding acquisition, J.-Y.J. All authors have read and agreed to the published version of the manuscript.

**Funding:** This research was supported by Korea Institute of Marine Science & Technology Promotion (KIMST) funded by the Ministry of Oceans and Fisheries (20210607).

**Institutional Review Board Statement:** Not applicable.

**Informed Consent Statement:** Not applicable.

**Data Availability Statement:** The data representing this study are available on request from the corresponding author. The data are not publicly available due to their containing information that could compromise the privacy of research participants.

**Conflicts of Interest:** The authors declare no conflict of interest.

**Appendix A**

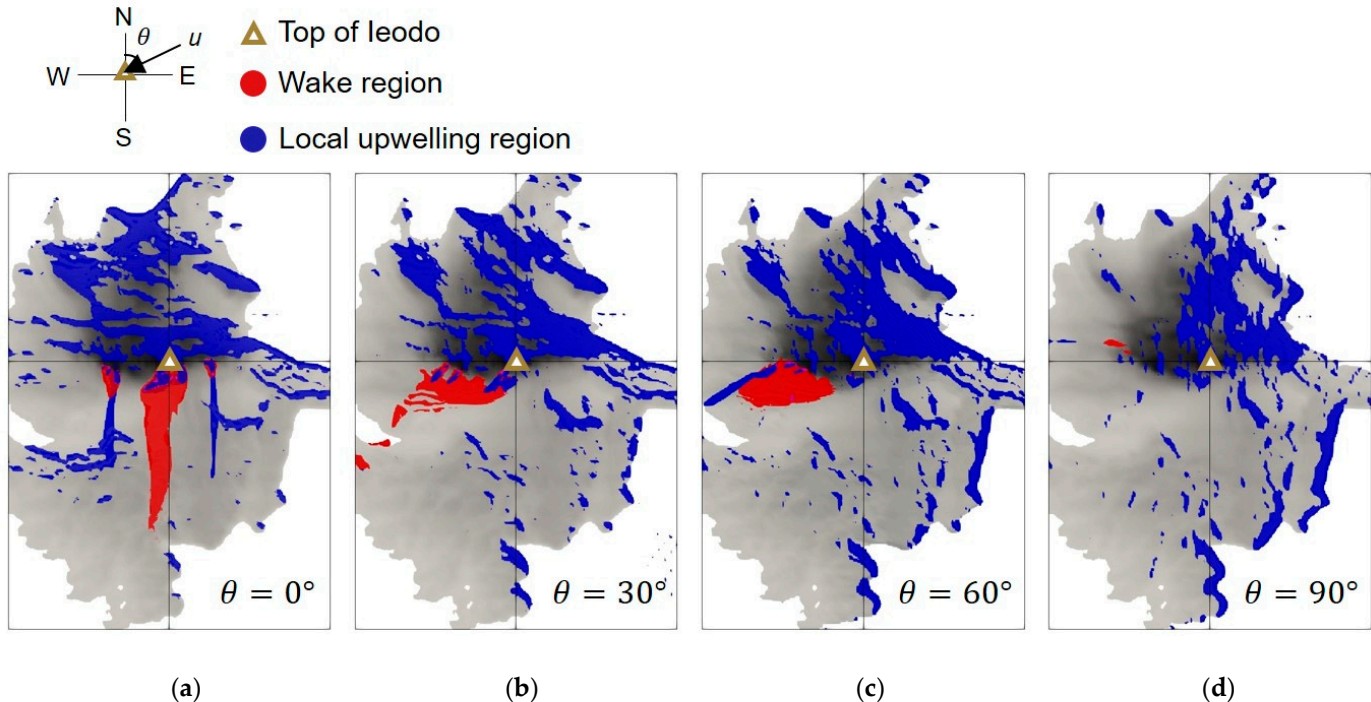

**Figure A1.** *Cont.*

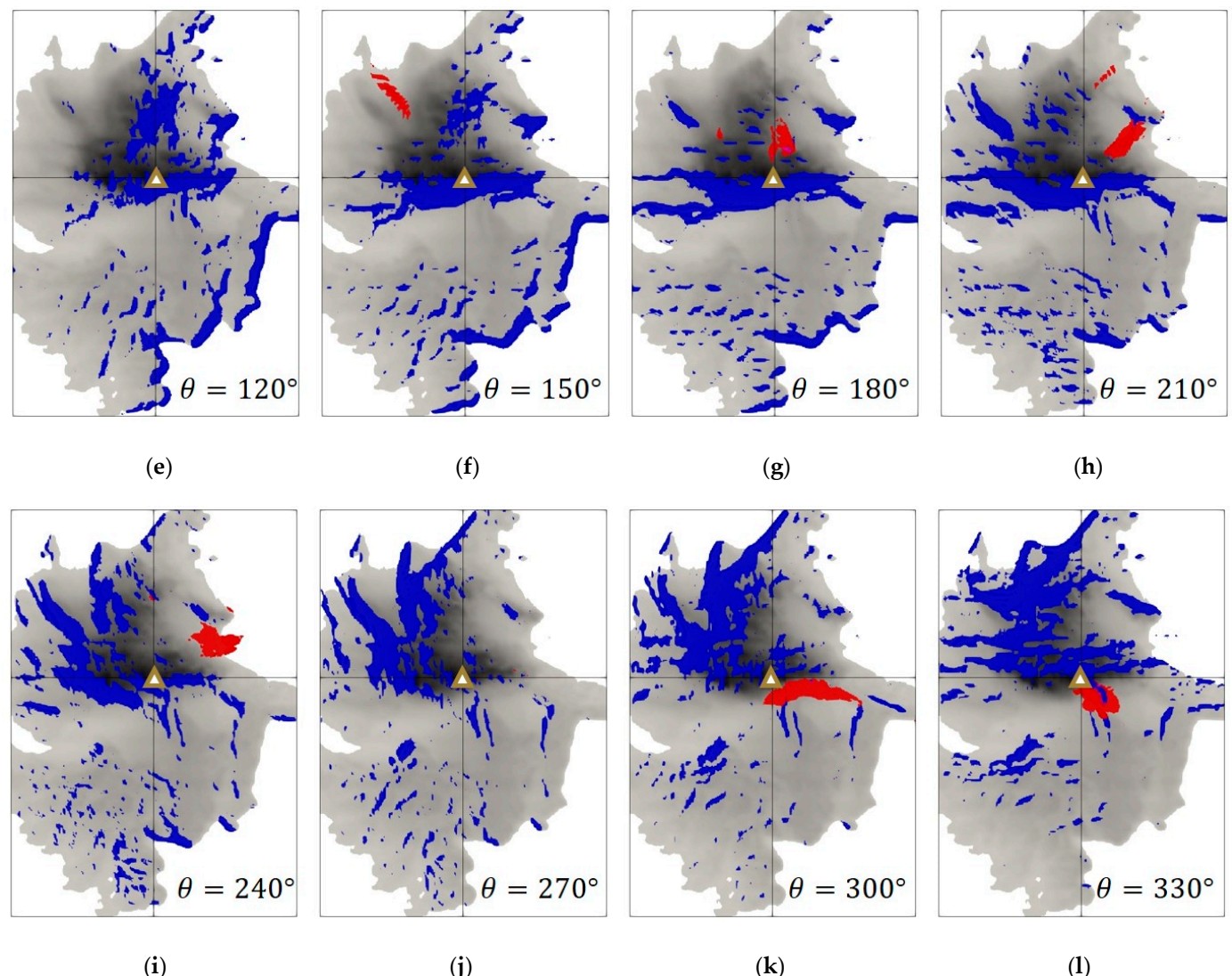

**Figure A1.** Results of numerical analysis of wake region and local upwelling volumes that occurs around the underwater reef (Ieodo) when the current direction changes at 30° intervals: (**a**) current direction is 0°; (**b**) 30°; (**c**) 60°; (**d**) 90°; (**e**) 120°; (**f**) 150°; (**g**) 180°; (**h**) 210°; (**i**) 240°; (**j**) 270°; (**k**) 300°; (**l**) 330°.

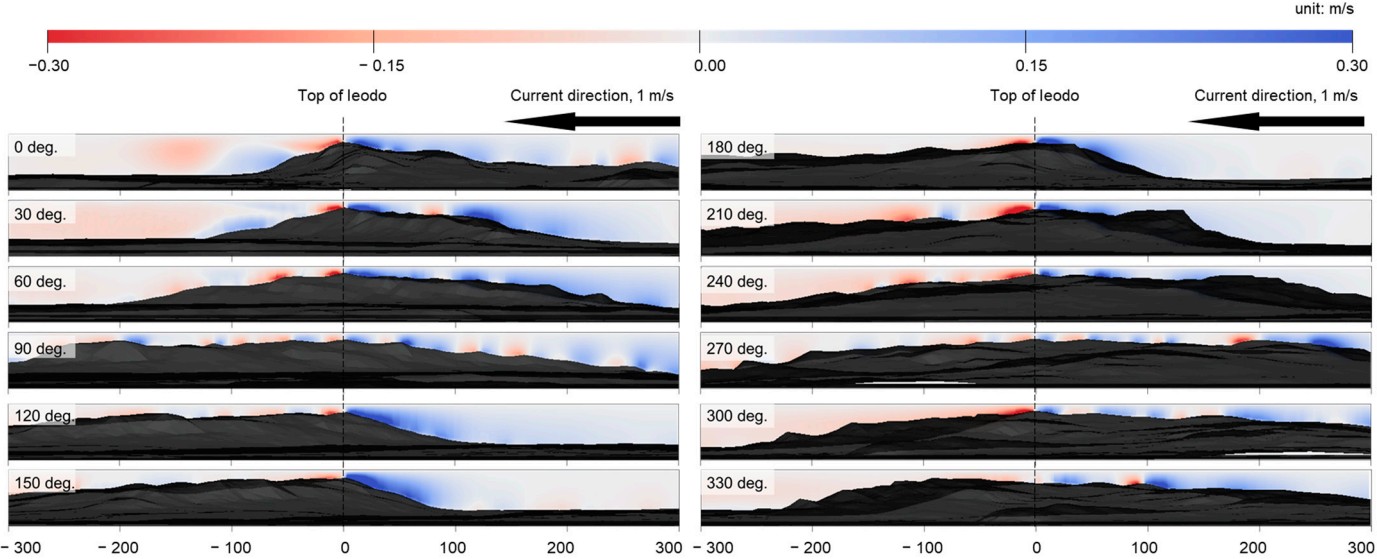

**Figure A2.** Results of numerical analysis of local upwelling velocity that occurs around the underwater reef (Ieodo) when the current direction changes at 30° intervals.

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
