# Peer review of "Identifying Hydraulic Characteristics Related to Fishery Activities Using Numerical Analysis and an Automatic Identification System of a Fishing Vessel"

_jmse, doi:10.3390/jmse10111619_

Round 1

Reviewer 1 Report

The approach provided is of interest and well conducted. The numerical simulations are quite interesting and well presented in the manuscript. Furthermore, the topic is of great relevance and a nice approach is provided to enhance the current knowledge and its applicability for different research fields. However, some issues could be enhanced prior its publication.

The authors provided a very nice result on Fig. 6 and generally in section 3.1, but a better description of the findings on how and why the local upwelling differs and how this affects would be desirable A short explanation is shown in the figure caption but not in the text, and the authors tested several conditions that should be informed. Also, it is noticed that the wake region and the local upwelling is mostly associated to the direction of 0°, but none is said why it does not appear in the other directions such as 120° and 270° where its indicated in Fig. 5 that they do not show an important wake region (not all readers know about the region).

For section 3.2, the results provided are very briefly mentioned (with just one paragraph) and mainly describing part of methods and part of what it is said in the figure caption of Fig. 7. It is expected to be enhanced this section, mainly providing to the reader with more references on the path of the FVI as a function of the time, otherwise the path looks like almost constant despite the change of the weak and upwelling regions. Moreover, section 3.2 should be enhanced to provide more than one paragraph and justify the need of the subtitle.

Furthermore, authors focus their results on volumetric information, and top view panels but it would be quite interesting to provide vertical slice plots showing the changes of the wake and upwelling regions, providing also valuable information gotten from the numerical model. Velocities changes are not provided (and which would also depend on the tides), and they are relevant as hydrodynamic properties obtained from the model. Authors are encouraged to provide these results. Finally results describing the effects of tides are missing. Therefore, they should be at least mentioned in results or in the discussion sections.

L169-178, L182-185, L188-193 are noticed that could be better integrated on the discussion section as they identify the needs of the research conducted but also comparison with further studies is provided and “discussed”. These lines could be considered as part of a more nutritive discussion than that currently provided.

Author Response

To reviewer #1

Thank you for reviewing the paper and leaving comments. Thank you for your comments, which could be made this paper better content. We respond to the comments as follows.

Comment #1 (C#1). The authors provided a very nice result on Fig. 6 and generally in section 3.1, but a better description of the findings on how and why the local upwelling differs and how this affects would be desirable A short explanation is shown in the figure caption but not in the text, and the authors tested several conditions that should be informed. Also, it is noticed that the wake region and the local upwelling is mostly associated to the direction of 0°, but none is said why it does not appear in the other directions such as 120° and 270° where its indicated in Fig. 5 that they do not show an important wake region (not all readers know about the region).

Answer #1 (A#1). Numerical analysis results of every 30° current directional and changes in current velocity with water depth are added to Appendix A, respectively. Based on the added analysis results, the changes in the wake region and local upwelling region according to the direction were described in section 3.1 as [L 222].

[L 222] The wake region usually occurs in conditions where an object blocks the flow. This is because the wake region occurs from pressure gradients and turbulence downstream of the object. This pattern can also be found in previous studies [10,37,38]. In the case of Ieodo, since the slope in the south direction is very steep, there is a large wake region in the 0° direction. On the other hand, in the 180° direction, the wake region is relatively small compared to 0°. This is because the terrain changes at the northern of the island is gentle. The local upwelling region is governed by the slope of the seabed topography. A large-scale local upwelling region develops when it flows along a steep slope of the seabed topography. Therefore, there is a characteristic that mainly appears upstream from the top of the Ieodo. As shown in Figure A1, it can be seen that the location where the local upwelling region and the wake region appear is spatially separated.

C#2. For section 3.2, the results provided are very briefly mentioned (with just one paragraph) and mainly describing part of methods and part of what it is said in the figure caption of Fig. 7. It is expected to be enhanced this section, mainly providing to the reader with more references on the path of the FVI as a function of the time, otherwise the path looks like almost constant despite the change of the weak and upwelling regions. Moreover, section 3.2 should be enhanced to provide more than one paragraph and justify the need of the subtitle.

A#2. Figure 7 shows the local upwelling region and the path of FV1 in time sequence. Chapter 3.2 has added [L 271] so that the title is enough descriptive.

[L 271] Table 1 shows the AIS information of FV1. At this time, the current direction could be determined through a numerical tidal current map. Thus, it can be checked whether there is a path of FV1 at the location where the wake region and the local upwelling region occur.

C#3. Furthermore, authors focus their results on volumetric information, and top view panels but it would be quite interesting to provide vertical slice plots showing the changes of the wake and upwelling regions, providing also valuable information gotten from the numerical model. Velocities changes are not provided (and which would also depend on the tides), and they are relevant as hydrodynamic properties obtained from the model. Authors are encouraged to provide these results. Finally results describing the effects of tides are missing. Therefore, they should be at least mentioned in results or in the discussion sections.

A#3. Analysis results according to the water depth are represented Figure A2. Kim (2019) confirmed that there was no significant effect on the development of the wake region and the local upwelling region according to the current velocity change through the flume experiment [22]. According to the flume experimental results, current velocity and wake region are inversely proportional, but their magnitude is insignificant. Therefore, in this study, the effect of the tidal difference was limited to the direction change.

C#4. L169-178, L182-185, and L188-193 are noticed that could be better integrated on the discussion section as they identify the needs of the research conducted but also comparison with further studies is provided and “discussed”. These lines could be considered as part of a more nutritive discussion than that currently provided.

A#4. L169-178, L182-185, and L188-193 have been moved to the discussion and reorganized.

Reviewer 2 Report

The title includes the words “Identifying Hydraulic Characteristics Related to Fish Attraction”. It is usually a qualitative statement. Although some relevant experimental studies have verified the preferences of fish for certain specific flow fields, the relevance of this statement is still ambiguous. Different species, even if the same species has different behavior characteristics in different life cycle stages. This paper does not verify this, but compares the data of numerical simulation and flow field investigation. Therefore, the title needs to be revised to highlight the focus of the article. If it is not modified, the predicted biomass of hydrodynamic spatial distribution waters must be added to further support the results.

Title 3.1, mumerical numerical

It is suggested to add pictures and detailed data of AIS, FV, etc. to show the experimental process.

Author Response

To reviewer #2

Thank you for reviewing the paper and leaving comments.

C#1. The title includes the words “Identifying Hydraulic Characteristics Related to Fish Attraction”. It is usually a qualitative statement. Although some relevant experimental studies have verified the preferences of fish for certain specific flow fields, the relevance of this statement is still ambiguous. Different species, even if the same species has different behavior characteristics in different life cycle stages. This paper does not verify this, but compares the data of numerical simulation and flow field investigation. Therefore, the title needs to be revised to highlight the focus of the article. If it is not modified, the predicted biomass of hydrodynamic spatial distribution waters must be added to further support the results.

A#1. We agree that there is still a lack of clear information about the hydrodynamic preferences of marine life. In this study, there is no data on the spatial distribution of biomass, life cycle, and behavior characteristics of marine lives, it is necessary to approach it more carefully. Therefore, the title has been modified as follows:

[Title] Identifying Hydraulic Characteristics Related to a Fishery Activities Using Numerical Analysis and an Automatic Identification System of a Fishing Vessel

C#2. Title 3.1, mumerical →numerical

A#2. A typo in Title 3.1 was corrected.

C#3. It is suggested to add pictures and detailed data of AIS, FV, etc. to show the experimental process.

A#3. The AIS information of FV1 and numerical analysis results are shown in addition to Table 1, Figure A1, and Figure A2 respectively.
